# Genotypic Effect on Olive (*Olea europaea*) Fruit Phenolic Profile

**DOI:** 10.3390/plants14131981

**Published:** 2025-06-28

**Authors:** Hande Yılmaz-Düzyaman, Lorenzo León, Raúl de la Rosa, Araceli Sánchez-Ortiz, Alicia Serrano, Francisco Luque, Carlos Sanz, Ana G. Perez

**Affiliations:** 1IFAPA Centro “Alameda del Obispo”, Avda. Menéndez Pidal s/n, 14004 Cordoba, Spain; lorenzo.leon@juntadeandalucia.es; 2Joint Doctoral Programme in Agricultural, Food, Forestry Engineering and Sustainable Rural Development, University of Córdoba and the University of Seville, 41005 Sevilla, Spain; 3Instituto de Agricultura Sostenible (CSIC), Avda, Menéndez Pidal s/n, 14004 Cordoba, Spain; raul.rosa@ias.csic.es; 4IFAPA, Centro Venta del Llano, Carr. Bailén-Motril, km. 18, 5, 23620 Mengíbar, Spain; araceli.sanchez.ortiz@juntadeandalucia.es; 5Departamento de Biología Experimental, Instituto Universitario de Investigación en Olivar y Aceites de Oliva, Universidad de Jaén, Campus Las Lagunillas s/n, 23071 Jaén, Spain; alserran@ujaen.es (A.S.); fjluque@ujaen.es (F.L.); 6Department of Biochemistry and Molecular Biology of Plant Products, Instituto de la Grasa, CSIC, Ctra. de Utrera, km. 1, 41013 Seville, Spain; carlos.sanz@ig.csic.es (C.S.); agracia@ig.csic.es (A.G.P.)

**Keywords:** *Olea europaea*, phenolic compounds, genetic variability, broad-sense heritability, olive breeding

## Abstract

Phenolic compounds are important targets in olive breeding due to their health benefits and impact on fruit and oil quality. Fruit phenolic profiling enables efficient screening of large germplasm collections without oil extraction, but environmental variability, especially year-to-year differences, affects their expression. The aim of this study was to assess the genotypic influence on fruit phenolic composition, based on a three-year evaluation of 10 wild olive genotypes and 75 cultivars from an olive core collection. Each genotype was sampled in at least two seasons, with 1 to 3 trees analyzed annually. Variance analysis revealed significant genetic variation among cultivars and notable genotype-by-year interactions for certain phenolics. Broad-sense heritability was generally high for most compounds, although some, such as ligstroside and ligstroside aglycone, showed greater environmental sensitivity. Best linear unbiased predictions (BLUPs) were highly correlated with average relative phenotypic values. Clustering analyses identified strong associations among key phenolic compounds and highlighted distinct metabolic profiles separating wild and cultivated genotypes, reflecting differences in phenolic accumulation patterns. These findings demonstrate the genetic and environmental influences on olive fruit phenolics and provide reliable estimates to support future marker-assisted selection studies aimed at developing useful tools in olive breeding programs.

## 1. Introduction

Virgin olive oil (VOO) is not only a staple of the Mediterranean diet but also one of the most valued vegetable oils due to its exceptional nutritional and health-promoting properties. It is widely acknowledged as a functional food, capable of exerting preventive effects against chronic diseases when consumed regularly [1,2]. Its unique chemical profile, markedly different from that of other vegetable oils, underpins its biological efficacy and sensory appeal. The health benefits of VOO are attributed to its high oleic acid content, which constitutes the majority of the saponifiable fraction, as well as to the unsaponifiable fraction, which is particularly rich in bioactive compounds [3]. Among these, phenolic compounds stand out for their strong antioxidant and anti-inflammatory activities, which are associated with the prevention of cardiovascular diseases, metabolic syndromes, and certain types of cancer [2,4].

In the olive fruit (*Olea europaea* L.), phenolic compounds fulfill essential roles in determining the quality and stability of olive-derived products, as well as contributing to plant defense mechanisms and resistance to pests and diseases. These molecules contribute to the fruit’s physiological processes and enhance its resilience to abiotic and biotic stressors, including drought, UV radiation, and pathogen attacks [5,6,7]. The primary classes of phenolics in olive fruit include secoiridoids, phenolic alcohols, and flavonoids. The most abundant and distinctive phenolic compounds found in the olive fruit are three secoiridoid glucosides: oleuropein and demethyloleuropein (both containing hydroxytyrosol) and ligstroside (containing tyrosol) [8,9,10]. These precursors, which are primarily responsible for the characteristic bitterness and pungency of VOO, are enzymatically hydrolyzed during fruit ripening and oil processing, resulting in the formation of the dialdehydic forms of decarboxymethyloleuropein and decarboxymethylligstroside aglycones (3,4-DHPEA-EDA and p-HPEA-EDA, respectively), as well as the aldehydic forms of oleuropein and ligstroside aglycones (3,4-DHPEA-EA and p-HPEA-EA, respectively) [11,12,13].

The final phenolic profile of VOO is shaped by both the initial composition in the fruit and enzymatic transformations occurring during malaxation [13,14]. Key enzymes, particularly olive β-glucosidase and polyphenol oxidase, play a central role in the hydrolysis, oxidation, and rearrangement of phenolic structures during oil extraction [15,16]. A strong correlation exists between the phenolic composition of the fruit and that of the resulting oil, particularly in terms of total phenolics (r = 0.66) and specific bioactive compounds, including oleuropein derivatives (r = 0.64), ligstroside derivatives (r = 0.73), and flavonoids (r = 0.74) [17]. However, the phenolic composition of olive fruit is known to be highly complex, shaped by genetic background [9,16,18], developmental stage [12,19,20], extraction process [21], and environmental conditions [22,23].

In recent years, significant progress has been made in characterizing phenotypic and genomic diversity within olive germplasm collections. The authors of Kaya [24] conducted a morphological evaluation of 183 olive accessions, focusing on traits such as leaf length, leaf width, fruit weight, stone weight, and the fruit flesh-to-pit ratio. Building upon this phenotypic foundation, Jiménez-Ruiz [25] re-sequenced 40 cultivated and 10 wild olive accessions to investigate genome-wide patterns of variation and shed light on the evolutionary dynamics underlying olive domestication. More recently, Moret [26] performed a genome-wide association study (GWAS) on fruit weight using the same set of 40 cultivated and 10 wild accessions, demonstrating the power of integrating phenotypic and genomic data, as well as the value of incorporating wild genotypes in genomic studies. Expanding this panel further, Moret [27] incorporated an additional 33 accessions to capture broader genetic diversity and, for the first time, identified a set of nine robust genetic markers validated for use in marker-assisted selection (MAS) during the early stages of olive breeding programs.

In this context, the present study provides high-resolution phenotypic data on phenolic traits from 75 cultivars and 10 wild genotypes—which overlap with the accessions sequenced by [27]—from the World Olive Germplasm Bank of Córdoba. Over a three-year period, phenolic profiles were monitored from multiple trees per genotype, enabling a robust assessment of temporal and genetic variation. A variance component analysis was performed to evaluate differences among genotypes, years, and their interaction, and broad-sense heritability estimates were calculated to quantify the genetic contribution to phenolic trait variation. Given the genotypic overlap with already-sequenced materials, this phenotypic dataset offers a valuable foundation for future genome-wide association studies (GWAS) aimed at identifying candidate genes regulating phenolic composition in olive fruit.

## 2. Results

Phenolic compounds were grouped into three main categories: hydroxytyrosol derivatives (DERHT), tyrosol derivatives (DERTY), and flavonoids (FLV), and their value expressed as a percentage of their correspondent main group (Table 1, Figure 1). A broad spectrum of variation was observed across the entire dataset for the oil phenolic components. A wide variability for total fruit phenol content was observed among cultivars, with average values ranging from 4390 in ‘Gordal Sevillana’ to 26,231 µg/g fruit in ‘Coratina’. Among the genotypes analyzed, seven wild olives showed remarkable higher total phenolic content than the cultivated varieties, with the highest concentration observed in the fruit of wild genotype ‘Aceb304’, averaging 74,999 µg/g. The mean total phenolic content was 15,412 μg/g. Among these, hydroxytyrosol derivatives were dominant, representing an average of 89.60% of the total phenolic content, with a large proportion (63.20%) consisting of oleuropein. While the FLV group displayed the highest variability among the three major groups, with a mean coefficient of variation of 52.18%, the most variable individual compounds were oleuropein aglycone (AGOLEU) and demethyloleuropein (DMOLEU), with mean coefficients of variation of 177.71% and 171.72%, respectively. Variability histograms showed distinctive skewed distributions for these compounds, with most of the values concentrated in the lower part of the graph, but still a widespread distribution for some genotypes.

The variance components extracted from the linear mixed-effects regression (lmer) model revealed that genotypic variance (σ_G_) was the predominant contributor to the total variance observed in the total phenols, as well as in the percentage of the three primary groups: DERHT, DERTY, and FLV. On the other hand, for OLEU, AGOLEU, LIGS, and AGLIGS, the genotype-by-year interaction (σ_GY_) was identified as the main contributing factor. The heritability (H^2^) predictions yielded the lowest values for AGLIGS and LIGS, while displaying significantly higher values for total phenols, the main groups, and several individual phenolic compounds (Table 2).

Best Linear Unbiased Predictions (BLUP) values extracted from the lmer model were used to perform hierarchical clustering heatmap using scaled data to visualize the associations among fruit phenolic components and olive genotypes (Figure 2). The heatmap reveals distinct clustering patterns both among genotypes and traits, indicating substantial phenotypic diversity in fruit phenolic composition across the evaluated genotypes. The genotypes were grouped into several clusters, suggesting underlying genetic similarities with respect to fruit phenolic traits. Among the 10 wild genotypes included in the study, 8 were clustered together based on their fruit phenolic composition, indicating a similar metabolic profile within this group. This cluster was primarily characterized by significantly elevated levels of total phenolics, the dimethyl forms of oleuropein and ligstroside, and tyrosol derivatives, while exhibiting comparatively low concentrations of hydroxytyrosol derivatives. The group comprising the other two wild genotypes also shared certain similarities with the main wild cluster, characterized by reduced relative levels of ligstroside and oleuropein, along with elevated relative levels of their metabolic derivatives, namely dimethylligstroside and dimethyloleuropein. Another particularly notable group was the final cluster, which stood out due to its elevated content of simple phenolic compounds, especially Hydroxytyrosol-1-O-glucoside, Hydroxytyrosol-4′-O-glucoside, and Tyrosol-1-O-glucoside.

## 3. Discussion

A wide variability was observed among all analyzed phenolic compounds, with the highest variation detected in AGOLEU, followed by DMOLEU, DMLIGS, and HT4G, respectively.

The linear mixed-effects model further revealed that genotypic variance (σ^2^_G_) was the principal source of total variability for most traits, especially for total phenols, DERHT, DERTY, and FLV, indicating strong genetic control. Total phenols and DERTY results for genotypic variance were in accordance with the previous findings [18,28]. However, genotype-by-year interaction (σ^2^_GY_) played a more significant role in compounds OLEU, AGOLEU, LIGS, and AGLIGS, suggesting that their expression is more sensitive to variation across harvesting seasons [18]. These results are in line with the calculated broad-sense heritability (H^2^) values, which were generally high for total phenols and most main groups (>0.90), but substantially lower for LIGS and AGLIGS, indicating a greater sensitivity to seasonal variation. Considering these findings, extending the evaluation across multiple years would be beneficial to enhance the robustness and precision of conclusions drawn for these specific compounds [29].

BLUPs provide unbiased estimates of the random effects associated with genotypes, allowing for a more accurate assessment of genetic potential. This analysis is particularly useful in unbalanced datasets, as it accounts for missing values and varying numbers of replicates per genotype. By estimating genetic values while minimizing environmental influences, BLUPs help in identifying superior genotypes for breeding and selection programs. Moreover, the resulting estimates can serve as reliable phenotypic inputs for downstream genetic analyses, such as genome-wide association studies (GWAS), thereby enhancing the identification of loci associated with key phenolic traits [27,30].

In the hierarchical clustering heatmap based on the BLUP values of fruit phenolic components, OLEU and LIGS were grouped within the same cluster and exhibited a strong positive association [28,31]. This co-clustering suggests that these two major secoiridoid compounds are regulated in a coordinated manner, possibly reflecting shared biosynthetic or biodegradation pathways across all genotypes [28,31]. Their positive correlation seems to be indicative of shared biosynthetic control or co-expression of key enzymes regulating their synthesis LIGS and OLEU are derivatives of elenolic acid linked to the phenolic alcohols tyrosol and hydroxytyrosol, which are generated from tyrosine and 3,4-dihydroxyphenylalanine (DOPA) through sequential processes including amino acid decarboxylation, primary amine oxidative deamination, and aldehyde reduction [32].

In contrast, DMOLEU and DMLIGS were clustered together in a separate group and displayed a negative correlation with OLEU and LIGS. This inverse relationship suggests a possible metabolic trade-off or regulatory shift between precursor and end-product forms within the secoiridoid biosynthesis. In this sense, although there are no conclusive studies, [11] always maintained that DMOLEU probably comes from oleuropein, and different studies have reported the interconversion of OLEU into DMOLEU along development and ripening of ‘Arbequina’ fruits [12,28]. These patterns are further supported by the broad-sense heritability (H^2^) estimates. DMOLEU and DMLIGS displayed relatively high heritability values (0.88 and 0.82, respectively), indicating that their variation is predominantly genetically controlled and stable across years. In contrast, OLEU and LIGS had notably lower heritability estimates (0.60 and 0.46, respectively), suggesting an influence of genotype-by-year interactions or year modulation in their accumulation.

Furthermore, a distinct clustering pattern was observed among eight wild olive genotypes (designated with the prefix ‘Aceb’), which grouped together based on their similar profiles. These genotypes exhibited high and positively correlated values (prominent red shades) for several key compounds, including DMOLEU, DMLIGS, AGOLEU, DERTY, and the total phenolic content (TOTAL). These patterns suggest that wild genotypes tend to accumulate higher levels of these specific phenolic constituents. The coordinated accumulation of these components in the wild types may reflect a shared biosynthetic or regulatory mechanism that promotes the production of specific phenolic subgroups in response to intrinsic genetic factors or ecological adaptation [6]. In contrast, although most wild genotypes contained nearly seven times higher levels of the major secoiridoids OLEU and LIGS compared to the cultivars, the contribution of these compounds to the total phenolic composition in wild genotypes was lower compared to cultivars (indicated by blue shades in the heatmap). This inverse relationship highlights a possible divergence in metabolic allocation between wild and domesticated olives. Similarly, a study analyzing other minor components of olive oils obtained from wild, crossed, and cultivated genotypes also reported significant differences in phenolic profiles between cultivated and wild olives, highlighting the impact of genetic background on the accumulation of these bioactive compounds [33]. Similar clustering patterns have been reported in genetic studies on olive domestication, where wild genotypes tend to group together, possibly reflecting ancestral metabolic traits [25].

In the cluster following the wild genotypes, a set of cultivated varieties, along with the remaining two wild genotypes, formed a separate cluster characterized by relatively low OLEU and LIGS content and higher levels of their metabolic derivatives DMOLEU and DMLIGS. These profiles, intermediate between wild and cultivated types, may suggest a conserved genomic or metabolic proximity likely shaped by shared ancestry or introgression events [25,34]. For instance, the cultivar ‘Dokkar’, classified within this cluster alongside wild types, has previously been linked to wild origins [34]. Moreover, Miho [7] demonstrated that resistant olive cultivars such as ‘Empeltre’ and ‘Frantoio’ activated the synthesis of aldehydic and demethylated phenolic forms during fruit ripening, which strongly inhibited *Colletotrichum* spore germination. In contrast, susceptible cultivars like ‘Hojiblanca’ and ‘Picudo’ predominantly accumulated hydroxytyrosol-4-O-glucoside—a simple phenol that showed no antifungal activity. Interestingly, in our study, the genotypes previously reported as resistant also tended to cluster together and were characterized by phenolic profiles enriched in bioactive compounds, whereas the susceptible ones clustered closely and were more associated with simple phenols.

In the third cluster group, the notably elevated levels of AGOLEU detected in five genotypes are of particular interest. AGOLEU is an aglycone primarily formed through the enzymatic hydrolysis of oleuropein during fruit crushing, rather than being naturally present under physiological conditions in intact olive fruit. Therefore, the presence of substantial amounts of AGOLEU in these genotypes may indicate genotype-specific metabolic differences that promote its formation. This observation suggests that AGOLEU could serve as a potential varietal marker, reflecting intrinsic biochemical traits—similar to the role of demethyloleuropein (DMOLEU) in the cultivar ‘Arbequina’, as previously described [12,28]. Moreover, consistent with earlier findings, a strong positive correlation was observed between DMOLEU and DMLIGS, supporting their coordinated accumulation within the phenolic biosynthetic pathway [17,18,31].

The following group contains the majority of the cultivars used in this study with low content of HT1G, HT4G, and TY1G. The fifth group demonstrates a trend close to the average for all phenolic components. And finally, the last cluster contains the double use or table olive cultivars with bigger or average fruit size and stands up with higher content of HT1G, HT4G, and TY1G.

Interestingly, the cluster closest to the wild genotypes predominantly includes olive oil cultivars, which typically have smaller fruits, producing oils with very high phenolic contents such as Dokkar, Menya, or Piñonera [28], whereas the most distant cluster is mainly composed of table olive cultivars characterized by normal to large fruit sizes.

## 4. Materials and Methods

### 4.1. Plant Materials

A total of 10 wild olive genotypes and 75 cultivars from the World Olive Germplasm Bank of Córdoba [35] were analyzed in this study. Samples were collected in mid-November over three consecutive years (2022–2024), with each cultivar having records spanning between 1 and 3 years. Within each year, samples were taken from 1 to 3 trees per genotype in mid-November. As a result, each genotype was analyzed in at least two of the three years; the dataset included between 2 and 9 data points per genotype, with a total of 442 data points per phenolic component.

### 4.2. Traits Evaluated

In this study, the main groups of fruit phenolic compounds analyzed were derivatives of hydroxytyrosol (DERHT), derivatives of tyrosol (DERTY), and flavonoids (FLV), along with total phenols. Additionally, specific individual compounds within each group were identified and evaluated. The DERHT group included hydroxytyrosol-1-O-glucoside (HT1G), hydroxytyrosol-4′-O-glucoside (HT4G), demethyloleuropein (DMOLEU), oleuropein (OLEU), oleuropein aglycone (AGOLEU), and verbascoside (VERBAS). In the DERTY group, the analyzed compounds were tyrosol-1-O-glucoside (TY1G), demethyl ligstroside (DMLIGS), ligstroside (LIGS), and ligstroside aglycone (AGLIGS). Meanwhile, the FLV group consisted of luteolin-7-O-glucoside (LUT7G), rutin (RUT), and apigenin-7-O-glucoside (API7G).

### 4.3. Extraction of Fruit Phenolic Compounds

Fruit phenolic compounds were isolated from each sample using a previously validated extraction protocol [16]. For the analysis of these compounds, precise longitudinal mesocarp slices (1 mm thick) were obtained from the pulp of approximately 20 fruits per genotype, with a total sample weight of about 1 g. The samples were subsequently placed in a dimethyl sulfoxide (DMSO) solution at a ratio of 6 mL per gram of fruit. This solution contained syringic acid (24 mg/mL) as an internal standard. The mixtures were then sonicated for 15 min and stored at 4 °C for 72 h to allow for extraction. Once the extraction was complete, the solutions were sonicated again and passed through a 0.45 μm nylon mesh for filtration. The filtered extracts were then stored at −20 °C until further analysis using High-Performance Liquid Chromatography (HPLC).

Phenolic extract analysis was performed using an HPLC system consisting of a Beckman Coulter liquid chromatography (Beckman Coulter, Brea, CA, USA) setup equipped with a System Gold 168 UV–Vis detector (Waters Corporation, Milford, MA, USA) set at 280 and 335 nm, an autosampler module 508, a Waters column heater maintained at a constant 35 °C, and a solvent module 126. Separation was carried out on a SunShell C18 column (4.6 × 250 mm, particle size 3.5 µm. ChromaNik Technologies Inc., Osaka, Japan). The mobile phase was a ternary mixture of 0.5% phosphoric acid (mobile phase A) and ethanol–acetonitrile (1:1) (mobile phase B). The following elution program was applied at a flow rate of 1 mL/min: 0–25 min, 5–30% B; 25–35 min, 30–38% B; 35–40 min, 38% B; 40–45 min, 38–45% B; 45–50 min, 45–100% B; 50–55 min, 100% B; 55–57 min, 100–5% B; and 57–60 min, 5% B.

The preliminary identification of compounds based on their UV–Vis spectra was validated through analysis with HPLC/ESI-qTOF-HRMS. The liquid chromatography system employed was the Dionex Ultimate 3000 RS UHPLC system (Thermo Fisher Scientific, Waltham, MA, USA), fitted with a column similar to the one used previously, but with 1% formic acid replacing 0.5% phosphoric acid in the mobile phase. A split post-column flow of 0.4 mL/min was directed straight into the electrospray ionization source of the mass spectrometer. The HPLC/ESIqTOF system performed mass analysis using a micrOTOF-QII High-Resolution Time-of-Flight mass spectrometer (UHRTOF) with qQ-TOF geometry (Bruker Daltonics, Bremen, Germany), which was equipped with an electrospray ionization (ESI) interface. Mass spectra were recorded in MS full scan mode, and the resulting data were analyzed using TargetAnalysis 1.2 software (Bruker Daltonics, Bremen, Germany).

### 4.4. Data Analysis

The content of individual phenolic compounds was expressed as a percentage of their respective main group, and each main group was expressed as a percentage of the total phenolic content. The mean values for each cultivar are provided in Appendix A. The total phenol content was retained in its original form (absolute concentrations, µg/g fresh weight), as it served as the basis for calculating both main groups and individual phenolics. All subsequent statistical analyses were performed using the derived percentage values of the main groups and their corresponding compounds. Descriptive statistics and histograms were generated for all main groups and their respective components. Variance analysis was performed using a linear mixed-effects regression (lmer) model on the unbalanced dataset of 85 genotypes, which contained missing values for certain trees or samples in some years. A total of 442 data points were collected for each phenolic component, across 14 subgroups, resulting in 6188 data points in the analysis.

Statistical analyses were conducted using R software (v4.3.2). The mixed-effects model analysis was performed using the lmer function from the lme4 package [36]. The model formula was specified as follows:parameter~(1|Year) + (1|Genotype) + (1|Genotype:Year) + (1|Replication)

In this model, parameter represents the dependent variable (phenolic components), while Year, Genotype, the interaction between Genotype:Year, and Replication were included as random effects. Random intercepts were added for Year, Genotype, and Replication, and the interaction between Genotype:Year was also treated as a random effect.

The variance components were calculated to assess the contributions of different factors to the total variance. The total variance was calculated by summing the genotypic variance (σ^2^_G_), year variance (σ^2^_Y_), genotype-by-year interaction variance (σ^2^_GY_), and residual variance (σ^2^ε). The proportion of total phenotypic variance that can be attributed to genetic factors, the broad-sense heritability (H^2^), was estimated using the following formula:H^2^ = σ^2^_G_/σ^2^_P_ = σ^2^_G_/[σ^2^_G_ + (σ^2^_GY_/Y) + (σ^2^ε/(Y × R))]
where σ^2^_G_ represents the genotypic variance and σ^2^_P_ phenotypic variance, σ^2^_GY_ corresponds to the variance attributed to the genotype-by-year interaction, and σ^2^ε denotes the residual variance [37]. Y indicates the total number of years considered in the study, while R refers to the mean number of replicates per genotype per year.

BLUPs (Best Linear Unbiased Predictions) values obtained from the percentage extracted from the lmer model by the ranef function. Heatmaps of BLUPs were generated using the ComplexHeatmap package in R [38]. This visualization method allows for an intuitive interpretation of genotype performance across different phenolic components, facilitating the identification of patterns and clustering among genotypes.

## 5. Conclusions

This study provides a comprehensive evaluation of the diversity and variation in fruit phenolic composition across a wide range of olive germplasm. The results emphasize the predominant role of genetic factors in determining fruit phenolic composition, particularly for traits such as total phenols, flavonoids, and certain secoiridoid derivatives. However, the notable genotype-by-year interaction observed for compounds like OLEU, AGOLEU, LIGS, and AGLIGS underscores the importance of multi-year evaluations for accurate phenotypic assessment. Clustering patterns based on phenolic profiles revealed distinct metabolic signatures associated with wild and cultivated genotypes, suggesting divergent evolutionary and/or selection pressures, particularly in the accumulation of secoiridoids and phenolic glycosides. The coordinated or inverse relationships between specific compound groups further point to underlying biosynthetic trade-offs and regulatory complexity. These findings offer valuable insights into the inheritance and metabolic regulation of phenolic compounds in olive fruit. Although we acknowledge that the phenolic content of olive oil is of primary importance for nutritional and commercial quality, it is well established that fruit phenolics are highly correlated with oil phenolics. Therefore, phenolic analyses in this study were performed on fruit tissue as a reliable proxy for oil phenolic composition. Moreover, the results obtained provide a robust framework for the identification of molecular markers and the development of marker-assisted selection strategies aimed at enhancing fruit quality and health-promoting traits in future olive breeding programs, particularly by using BLUP estimates as reliable input data for genome-wide association studies (GWAS).

## Figures and Tables

**Figure 1 plants-14-01981-f001:**
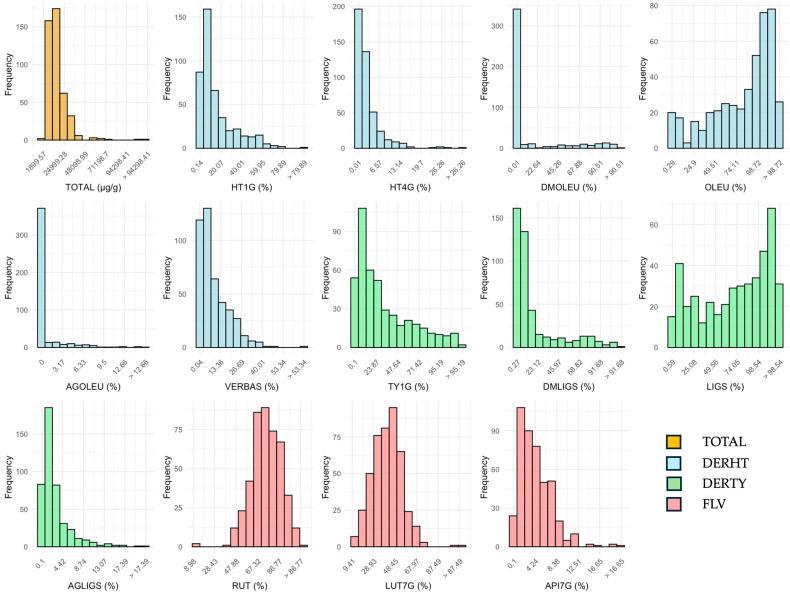
Variability histograms for total (TOTAL) and individual phenols in olive fruit: HT1G: hydroxytyrosol-1-O-glucoside, HT4G: hydroxytyrosol-4′-O-glucoside, DMOLEU: demethyloleuropein, OLEU: oleuropein, AGOLEU: oleuropein aglycone, VERBAS: verbascoside, TY1G: tyrosol-1-O-glucoside, DMLIGS: demethylligstroside, LIGS: ligstroside, AGLIGS: ligstroside aglycone, RUT: rutin, LUT7G: luteolin-7-O-glucoside, API7G: apigenin-7-O-glucoside. Individual phenols content is represented as percentage of their correspondent main group: hydroxytyrosol derivatives: DERHT; tyrosol derivatives: DERTY; and flavonoids: FLV (different colors represent the groups to which individual components belong).

**Figure 2 plants-14-01981-f002:**
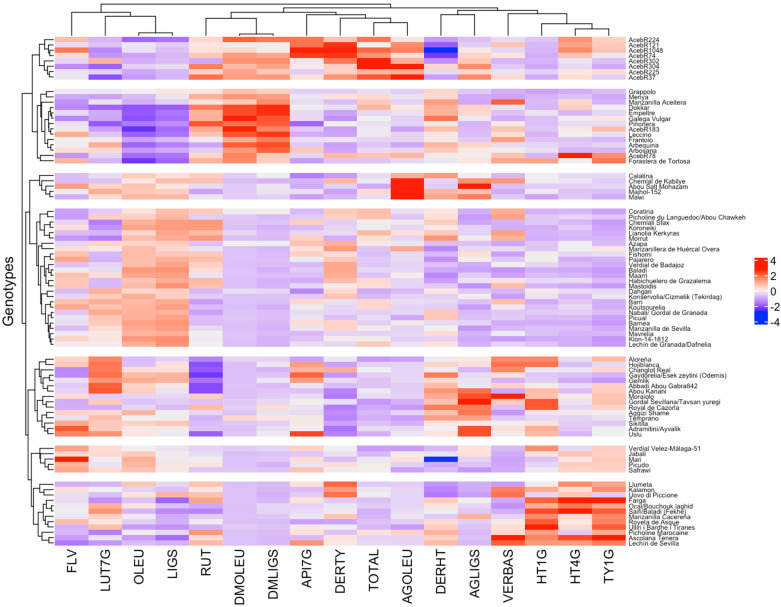
Hierarchical clustering heatmap of BLUP values for fruit phenolic components. TOTAL: total phenols, DERHT: hydroxytyrosol derivatives, DERTY: tyrosol derivatives, FLV: flavonoids, HT1G: hydroxytyrosol-1-O-glucoside, HT4G: hydroxytyrosol-4′-O-glucoside, DMOLEU: demethyloleuropein, OLEU: oleuropein, AGOLEU: oleuropein aglycone, VERBAS: verbascoside, TY1G: tyrosol-1-O-glucoside, DMLIGS: demethylligstroside, LIGS: ligstroside, AGLIGS: ligstroside aglycone, RUT: rutin, LUT7G: luteolin-7-O-glucoside, API7G: apigenin-7-O-glucoside.

**Table 1 plants-14-01981-t001:** Mean, coefficient of variation (CV), minimum and maximum values for each phenolic group (the content of individual phenols is expressed as a percentage of their corresponding main group, and the main groups are expressed as a percentage of total phenols).

Parameter	Mean	CV (%)	Min	Max
Total phenols (μg/g)	15,412.15	74.30	4390.84	74,999.68
DERHT (%)	89.60	3.96	76.70	96.28
HT1G	12.29	90.61	0.54	49.78
HT4G	2.56	114.70	0.07	15.86
DMOLEU	13.78	171.72	0.04	80.21
OLEU	63.20	39.59	0.70	97.86
AGOLEU	0.89	177.71	0.03	6.71
VERBAS	7.29	84.62	0.22	25.58
DERTY (%)	6.28	48.12	1.51	14.71
TY1G	26.71	76.10	1.74	87.96
DMLIGS	15.75	117.90	1.10	66.36
LIGS	55.24	48.06	5.42	96.48
AGLIGS	2.30	64.28	0.58	6.77
FLV (%)	4.13	52.18	0.78	14.55
RUT	62.42	15.34	39.64	84.17
LUT7G	34.09	28.35	10.88	53.49
API7G	3.50	61.77	1.02	15.99

DERHT: hydroxytyrosol derivatives, DERTY: tyrosol derivatives, FLV: flavonoids, HT1G: hydroxytyrosol-1-O-glucoside, HT4G: hydroxytyrosol-4′-O-glucoside, DMOLEU: demethyloleuropein, OLEU: oleuropein, AGOLEU: oleuropein aglycone, VERBAS: verbascoside, TY1G: tyrosol-1-O-glucoside, DMLIGS: demethylligstroside, LIGS: ligstroside, AGLIGS: ligstroside aglycone, RUT: rutin, LUT7G: luteolin-7-O-glucoside, API7G: apigenin-7-O-glucoside.

**Table 2 plants-14-01981-t002:** The contribution of each variance component (%)—genotypic: σ_G_, year: σ_Y_, genotype-by-year interaction: σ_GY,_ and error: σ_E_—to the total variance calculated based on the results of the linear mixed-effects regression (lmer) and broad-sense heritability (H^2^).

Component	σ_G_	σ_Y_	σ_GY_	σ_E_	H^2^
Total phenols	71.94	10.50	11.06	6.50	0.94
DERHT	72.11	10.27	10.73	6.89	0.94
HT1G	43.92	10.65	23.62	21.81	0.78
HT4G	79.63	2.12	9.85	8.41	0.94
DMOLEU	66.62	3.99	22.03	7.36	0.88
OLEU	22.76	27.21	40.10	9.92	0.60
AGOLEU	29.61	26.66	39.95	3.78	0.68
VERBAS	64.96	0.66	11.23	23.15	0.89
DERTY	75.35	7.71	11.23	5.72	0.94
TY1G	76.69	2.68	14.48	6.15	0.93
DMLIGS	59.22	1.16	36.67	2.95	0.82
LIGS	15.79	23.44	50.63	10.14	0.46
AGLIGS	7.30	29.05	61.01	2.64	0.26
FLV	73.51	4.06	4.71	17.72	0.94
RUT	77.36	4.00	4.88	13.76	0.95
LUT7G	63.44	1.61	2.78	32.18	0.90
API7G	76.29	8.89	11.17	3.64	0.95

DERHT: hydroxytyrosol derivatives, DERTY: tyrosol derivatives, FLV: flavonoids, HT1G: hydroxytyrosol-1-O-glucoside, HT4G: hydroxytyrosol-4′-O-glucoside, DMOLEU: demethyloleuropein, OLEU: oleuropein, AGOLEU: oleuropein aglycone, VERBAS: verbascoside, TY1G: tyrosol-1-O-glucoside, DMLIGS: demethylligstroside, LIGS: ligstroside, AGLIGS: ligstroside aglycone, RUT: rutin, LUT7G: luteolin-7-O-glucoside, API7G: apigenin-7-O-glucoside.

## Data Availability

The data presented in this study are available on request from the corresponding author.

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
