# Peer review of "Genotypic Effect on Olive (*Olea europaea*) Fruit Phenolic Profile"

_plants, 2025, doi:10.3390/plants14131981_

Round 1
Reviewer 1 Report
Comments and Suggestions for Authors
This is a sollid contribution for future bedding programs; the cluster anaylses between wild and cultivated genotypes is particularly interesting.
My only comment is that 1 to 3 trees for 2 seasons is a limited sample for identifying environmental effects.
Reviewer 2 Report
Comments and Suggestions for Authors
The present study provides phenotypic data on phenolic characteristics of 75 olive cultivars and 10 wild olive genotypes in Spain.
The introduction provides an acceptable details on previous knowledge regarding olive oil, phenolic profile of olive fruit and oil. The experimental approaches and data analyses are acceptable, employing suitable methods to achieve the study’s objectives.
The results contain some useful results such as i) variance analysis demonstrated significant genetic variation among cultivars ii) broad-sense heritability was generally high for most measured phenolic compounds iii) BLUPs and phenotypic and genotypic correlations showed strong associations among phenolic compounds iv) cluster analyses highlighted distinct metabolic profiles separating wild and cultivated genotypes.
The discussion section provide an acceptable comparison with the previous studies. Authors provided a suitable conclusion. More details are needed on potential future research directions and on the study’s limitations of the study.
Amount of cited literatures are acceptable and the format of references are generally suit to journal requirements. L390-391: Why the title is with captital letters?
This study includes some interesting elements that could make it suitable for publication after minor revisions.
Other Suggestions and Comments:
L135: The percentage …
Figure 2: The letter size needs to be increased in order to increase visibility.
L371 and L419 and L446 and L449: Olea europaea – should be in italic
Reviewer 3 Report
Comments and Suggestions for Authors
The aim of the study is to investigate the genetic and environmental influences on olive fruit phenolics.
The work is well structured and discussed.
Innovative statistic tools, such as BLUP, are used.
Results are consistent with conclusions.
References are well placed.
However, in order to improve the quality of manuscript I suggest the following revisions:
- In the ‘Abstract’ please specify the reason of the research
- In the ‘Introduction section’,
- Please explain better the differences both in terms of methods and results between this work and that carried out by the same group in https://www.mdpi.com/2311-7524/9/10/1087
Some minor issues:
- Please The format of the citations at Lines 68, 71 and 73 is not correct. Please modify.
Reviewer 4 Report
Comments and Suggestions for Authors
I recommend that the authors consider my comments and suggestions to maximise the scientific value of their excellent findings.
My suggestions for improving the paper can be found embedded as comments within the PDF file.
Furthermore, it is essential that the authors provide the metadata (absolute and relative concentrations for each variety, including means and standard deviations) to ensure reproducibility and to maximise the scientific impact of the paper.

Author Response
Please see the attachment.
The mean of each cultivar (absolute value) is added as a Supplementary Table 1.
